# Multi-Objective SPIBB: Seldonian Offline Policy Improvement with Safety Constraints in Finite MDPs

**Harsh Satija**
McGill University, Mila
`harsh.satija@mail.mcgill.ca`

**Philip S. Thomas**
University of Massachusetts
`pthomas@cs.umass.edu`

**Joelle Pineau**
McGill University, Mila, Facebook AI Research
`jpineau@cs.mcgill.ca`

**Romain Laroche**
Microsoft Research
`romain.laroche@microsoft.com`

## Abstract

We study the problem of Safe Policy Improvement (SPI) under constraints in the offline Reinforcement Learning (RL) setting. We consider the scenario where: (i) we have a dataset collected under a known baseline policy, (ii) multiple reward signals are received from the environment inducing as many objectives to optimize. We present an SPI formulation for this RL setting that takes into account the preferences of the algorithm's user for handling the trade-offs for different reward signals while ensuring that the new policy performs at least as well as the baseline policy along each individual objective. We build on traditional SPI algorithms and propose a novel method based on Safe Policy Iteration with Baseline Bootstrapping (SPIBB, Laroche et al., 2019) that provides high probability guarantees on the performance of the agent in the true environment. We show the effectiveness of our method on a synthetic grid-world safety task as well as in a real-world critical care context to learn a policy for the administration of IV fluids and vasopressors to treat sepsis.

## 1 Introduction

Reinforcement Learning (RL) as a paradigm for sequential decision-making (Sutton, 1988) has shown tremendous success in a variety of simulated domains (Mnih et al., 2015; Silver et al., 2017; OpenAI, 2018). However, there are still quite a few challenges between the traditional RL research and real-world tasks. Most of these challenges stem from assumptions that are rarely satisfied in practice (Dulac-Arnold et al., 2019), or the inability of the algorithm's user to specify the desired behavior of the agent without being a domain expert (Thomas et al., 2019). We focus on the real-world application point of view and posit the following requirements:

- **Multiple reward functions:** Traditional RL methods assume a single scalar reward is present in the environment. However, most real-world tasks, have multiple (possibly conflicting) objectives or constraints that need to be taken into consideration together, such as the signals related to the safety (physical well-being of the agent or the environment), budget utilization (energy or maintenance costs), etc.

- **Stakeholder control of the trade-off:** The ML practitioners should have the ability to control the different trade-offs the agent is making and choose the one they consider best for the task at hand.

- **Offline setting:** In many real-world domains (e.g., healthcare, finance or autonomous vehicles), there is an abundance of data, collected under a sub-optimal policy, but training the agent directly via interactions with the environment is expensive and risky. We assume that we only have access to a dataset of past trajectories that can be used for training (Lange et al., 2012).

35th Conference on Neural Information Processing Systems (NeurIPS 2021).

- **Preventing unintended behavior:** We want the agent to be robust to both extrapolation errors from offline RL and misaligned objectives that are poor proxy of the user's intentions and algorithm's actual performance (Ng et al., 1999; Amodei et al., 2016). We consider the case where the user can specify undesirable behavior in the context of the performance observed in the batch.
- **Practical guarantees:** We want guarantees about the undesirable behavior that might be caused by the agent in the real-world. We care about the results that can be obtained using the finite amount of samples we have in the batch, and aim to provide some measure of confidence in deploying the agents in the environment.

To achieve this set of properties, we adopt the Seldonian framework (Thomas et al., 2019), which is a general algorithm design framework that allows high-confidence guarantees for constraint satisfaction in a multi-objective setting. Based on the above specifications, we seek to answer the question: *if we are given a batch of data collected under some (suboptimal) behavioral policy and some user preference, can we build a policy improvement algorithm that returns a policy with practical high-confidence guarantees on the performance of the policy w.r.t. the behavioral policy?*

We acknowledge that there are other important challenges in RL, such as partial observability, safe exploration, non-stationary environments and function approximation in high-dimensional spaces, that also stand in the way of making RL a more applicable paradigm. These challenges are beyond the scope of this work, which should rather be thought of as taking a step towards this broader goal.

In Section 2, we present our contribution positioned with respect to other related work. In Section 3, we formalize the setting and then extend traditional SPI algorithms to this setting. We then show it is possible to extend the previous work on Safe Policy Iteration (SPI), particularly Safe Policy Iteration with Baseline Bootstrapping (SPIBB, Laroche et al., 2019), for the design of agents that satisfy the above requirements. We show that the resulting algorithm is theoretically-grounded and provides practical high-probability guarantees. We extensively test our approach on a synthetic safety-gridworld task in Section 4 and show that the proposed algorithm achieves better data efficiency than the existing approaches. Finally, we show its benefits on a critical-care task in Section 5. The accompanying codebase is available at `https://github.com/hercky/mo-spibb-codebase`.

## 2 Related work

**Multi-Objective RL (MORL):** Traditional multi-objective approaches (Mannor and Shimkin, 2004; Roijers et al., 2013; Liu et al., 2014) focus on finding the Pareto-frontier of optimal reward functions that gives all the possible trade-offs between different objectives. The user can then select a policy from the solution set according to their arbitrary preferences. In practice, an alternate trial and error based approach of scalarization is used to transform the multiple reward functions into a scalar reward based on preferences across objectives (usually, by taking a linear combination). Most traditional MORL approaches have focused on the online, interactive settings where the agent has access to the environment. While some recent approaches are based on off-policy learning methods (Lizotte et al., 2012; Van Moffaert and Nowé, 2014; Yang et al., 2019; Abdolmaleki et al., 2020), they lack guarantees. In contrast, our work focuses exclusively on learning in the offline setting and gives high-probability guarantees on the performance in the environment.

**Constrained-RL:** RL under constraints frameworks, such as Constrained MDPs (CMDPs, Altman, 1999), present an alternative way to define preferences in the form of constraints over policy's returns. Here, the user assigns a single reward function to be the primary objective (to maximize) and hard constraints are specified for the others. The major limitation of this setting is that it assumes the thresholds for the constraints are known a priori. Le et al. (2019) study offline policy learning under constraints and provide performance guarantees w.r.t. the optimal policy, but their work relies on the concentrability assumption (Munos, 2003).

Concentrability is a strong assumption that upper bounds the ratio between the future state-action distributions of any non-stationary policy and the baseline policy under which the dataset was generated by some constant. From a practical perspective, it is unclear how to get a tractable estimate of this constant, as the space of future state-action distributions of non-stationary policies is vast. Thus, this constant can be arbitrarily huge, potentially even infinite when the baseline policy fails to cover the support of the space of all non-stationary policies (such as in the low-data regime), leading to the performance bounds given by these methods to blow up (and even be unbounded). Additionally, the guarantees in Le et al. (2019) are only valid with respect to the performance of the optimal policy.

In this work, we instead focus on the performance guarantees based on returns observed in the dataset, as it does not require making any of the above assumptions.

**Reward design:** Reward-design (Sorg et al., 2010) and reward-modelling approaches (Christiano et al., 2017; Littman et al., 2017; Leike et al., 2018) focus on designing suitable reward functions that are consistent with the user's intentions. These approaches rely heavily on the human or simulator feedback, and thus do not carry over easily to the offline setting.

**Seldonian-RL (and Safe Policy Improvement):** The Seldonian framework (Thomas et al., 2019) is a general algorithm design framework that allows the user to design ML algorithms that can avoid undesirable behavior with high-probability guarantees. In the context of RL, the Seldonian framework allows to design policy optimization problems with multiple constraints, where the solution policies satisfy the constraints with high-probability. In the offline-RL setting, SPI refers to the objective of guaranteeing a performance improvement over the baseline with high-probability guarantees (Thomas et al., 2015a; Petrik et al., 2016; Laroche et al., 2019). Therefore, SPI algorithms are a specific setting that falls in the general category of Seldonian-RL algorithms.

We focus on two categories of SPI algorithms that provide practical error bounds on safety: SPIBB (Laroche et al., 2019) that provides Bayesian bounds and HCPI (Thomas et al., 2015a,b) that provides frequentist bounds. SPIBB methods constrain the change in the policy according to the local model uncertainty. SPIBB has been formulated in the context of a single reward function, and as such does not handle multiple rewards and by extension also lacks the ability for the user to specify preferences. Our primary focus is to provide a construction for extending the SPIBB methodology to the multi-objective setting that handles user preferences and provides high-probability guarantees.

Instead of relying on model uncertainty, HCPI methods utilize the high-confidence lower bounds on the Importance Sampling (IS) estimates of a target policy's performance to ensure safety guarantees. HCPI has been applied to solve Seldonian optimization problems for constrained-RL setting using an enumerable policy class. Thomas et al. (2019) suggested using HCPI for the MORL setting, and we build on that idea. Particularly, we show how HCPI can be implemented with stochastic policies in the context of our setting with user preferences and baseline constraints.

## 3 Methodology

### 3.1 Setting

We consider the setting where the agent's interactions with the environment can be modelled as a Markov Decision Process (MDP, Bellman, 1957). Let $\mathcal{X}$ and $\mathcal{A}$ respectively be the (finite) state and action spaces. Let $p^\star : \mathcal{X} \times \mathcal{A} \to \mathscr{P}(\mathcal{X})$ denote the true (unknown) transition probability function, where $\mathscr{P}(\mathcal{X})$ denotes the set of probability distributions on $\mathcal{X}$. Without loss of generality, we assume that the process deterministically begins in the state $x_0$. We define $[N]$ to be the set $\{0, 1, \ldots, N-1\}$ for any positive integer $N$. Let there be $d$ different reward signals and $\boldsymbol{r}^\star = \{r_k\}_{k \in [d]} : \mathcal{X} \times \mathcal{A} \to [-r_\top, r_\top]^d$ be the true (unknown) stochastic multi-reward signal.[1] Finally, $\boldsymbol{\gamma} = \{\gamma_k\}_{k \in [d]} \in [0,1)^d$ is the multi-discount-factor.

The MDP, $m^\star$, can now be defined with the tuple $(\mathcal{X}, \mathcal{A}, p^\star, \boldsymbol{r}^\star, \boldsymbol{\gamma}, x_0)$. A policy $\pi : \mathcal{X} \to \mathscr{P}(\mathcal{A})$ maps a state to a distribution over actions. We denote by $\Pi$ the set of stochastic policies. We consider the infinite horizon discounted return setting. For any $k \in [d]$, the $k^{\text{th}}$ reward value function $v_{m,k}^\pi(x) : \mathcal{X} \to \mathbb{R}$ denotes the expected discounted sum of rewards when when following policy $\pi$ in an MDP $m$ starting from state $x$. Analogously, we define the state-action value functions for performing action $a$ in state $x$ in MDP $m$ under $\pi$ for rewards as $q_{m,k}^\pi(x,a)$. Let $\text{Adv}_{m,k}^\pi(x,a) = q_{m,k}^\pi(x,a) - v_{m,k}^\pi(x)$ denote the corresponding advantage function. The expected return of policy $\pi$ w.r.t. the $k^{\text{th}}$ reward in the true MDP $m^\star$ is denoted by $\mathcal{J}_{m^\star,k}^\pi = v_{m^\star,k}^\pi(x_0) = \mathbb{E}_{\pi,m^\star}[\sum_{t=0}^\infty \gamma_k^t R_{k,t} \mid X_0 = x_0]$, where action $A_t \sim \pi(\cdot \mid X_t)$, immediate reward $R_{k,t} \sim r_k^\star(\cdot \mid X_t, A_t)$, and state $X_{t+1} \sim p^\star(\cdot \mid X_t, A_t)$.

We consider the offline setting, where instead of having access to the environment we have a pre-collected dataset of trajectories denoted by $\mathcal{D} = \{\tau_i\}_{i \in [|\mathcal{D}|]}$, where $|\mathcal{D}|$ denotes the number of trajectories in the dataset. A trajectory $\tau$ of length $T$ is an ordered set of transition tuples of the form $\tau = \{x_i, a_i, x_i', \boldsymbol{r}_i\}_{i \in [T]}$, where $x_i'$ denotes the state at the next time-step. We denote the Maximum

---

[1]Costs, which are meant to be minimized, can be expressed as negative rewards.

Likelihood Estimation (MLE) of the MDP with $\hat{m} = (\mathcal{X}, \mathcal{A}, \hat{p}, \hat{\boldsymbol{r}}, \boldsymbol{\gamma}, x_0)$, where $\hat{p}$ and $\hat{\boldsymbol{r}}$ denote the transition and reward models estimated from the dataset's statistics.

**Assumption 3.1** (Baseline policy). We assume that we have access to the policy that generated the dataset. We call such policy the baseline policy and denote it by $\pi_b$. [2]

### 3.2 Problem formulation

We consider safe policy improvement with respect to the baseline according to the $d$ dimensions of the multi-objective setting. Therefore, under a Bayesian approach, we search for target policies such that they perform better (up to a precision error $\zeta$) than the baseline along every objective function with high probability $1 - \delta$, where $\zeta$ and $\delta$ are hyper-parameters controlled by the user, denoting the risk that the practitioner is willing to take. We denote by $\Pi_A$ the set of admissible policies that satisfy:

$$\mathbb{P}\left(\forall k \in [d], \mathcal{J}_{m^\star, k}^{\pi} - \mathcal{J}_{m^\star, k}^{\pi_b} > -\zeta \Big| \mathcal{D}\right) > 1 - \delta. \tag{1}$$

In the multi-objective case, there does not exist a single optimal value, but a Pareto frontier of optimal values. One way to evaluate the MORL problems is via the *multiple-policy* approaches (Vamplew et al., 2011; Roijers et al., 2013) that compute the policies that approximate the true optimal Pareto-frontier. However, note that optimality and safety are contradicting objectives. It is not clear how (and if) one can make claims about optimality in the offline setting without bringing in additional unrealistic assumptions (Section 2, MORL). Instead, we take an alternate approach inspired by another category of MORL methods called *single-policy* (Roijers et al., 2013; Van Moffaert and Nowé, 2014) where the trade-offs between different objectives are explicitly controlled by the user via providing a scalarization or preferences over objectives. We assume the user preference $\boldsymbol{\lambda} = \{\lambda_k\}_{k \in [d]}$ is given as an input to our algorithms, and is used for scalarization of the objectives, where $\lambda_k \in \mathbb{R}^+$. Our objective therefore becomes

$$\arg\max_{\pi \in \Pi_A} \sum_{k \in [d]} \lambda_k \mathcal{J}_{m^\star, k}^{\pi}. \tag{2}$$

The above formulation gives freedom to the user in terms of what particular quantity they want to optimize via $\boldsymbol{\lambda}$, but still ensures that the solution policy performs as well as the baseline policy across all $d$ objectives. Note that our explicit goal is to maximize the objective specified by the user. However, the user might make mistakes in specifying this objective (Section 2, Reward design), and the above formulation offers guarantees that prevent deteriorating the performance of the policy across any of the $d$ objectives. This allows the user to to experiment with different reward design strategies in safety-critical settings without worrying about the risks of ill-defined scalarizations. A naïve approach would be applying the user scalarization to also define the safety constraints. However, this construction fails to prevent undesirable behavior for the individual objectives (shown in Appendix A).

### 3.3 Multi-Objective SPIBB (MO-SPIBB)

Robust MDPs (Iyengar, 2005; Nilim and El Ghaoui, 2005) can be regarded as an approximation of the Bayesian formulation by partitioning the MDP space $\mathcal{M}$ into two subsets: the subset of plausible MDPs $\Xi$ and the subset of implausible MDPs. The plausible set is classically constructed from concentration bounds over the reward and transition function:

$$\Xi = \left\{ m, \text{ s.t. } \forall x, a, \begin{array}{l} \|p(\cdot|x, a) - \hat{p}(\cdot|x, a)\|_1 \le e(x, a), \\ \|\boldsymbol{r}(x, a) - \hat{\boldsymbol{r}}(x, a)\|_\infty \le e(x, a)r_\top \end{array} \right\},$$

where $e$ is an upper bound on the state-action error function of the model that are classically obtained with concentration bounds, such that the true environment $m^\star \in \Xi$ with high probability $1 - \delta$. In the single objective framework, Laroche et al. (2019) empirically show that optimising the worst-case performance policy in $\Xi$ provides policies that are too conservative. Petrik et al. (2016) prove that it is NP-hard to find the policy $\pi$ that maximises the worst-case policy improvement over $\Xi$.

Instead, the SPIBB methodology (Laroche et al., 2019) consists in searching for a policy that maximizes the safe policy improvement in the MLE MDP, under some policy constraints: SPIBB

---

[2]Simão et al. (2020) proved that SPIBB/Soft-SPIBB bounds may be obtained with an estimate of $\pi_b$.

and Soft-SPIBB (Nadjahi et al., 2019) policy search constraints both revolve around the idea that we must only consider policies for which the policy improvement may be accurately estimated. Using $\pi_b$ as reference, SPIBB allows policy changes only in state-action pairs for which more than $n_\wedge$ samples have been collected. Soft-SPIBB extends this by applying soft constraints that allow slight changes in the policy for the uncertain state-action pairs, which are controlled by an error bound related to model uncertainty. As such, on low-confidence transitions, this class of methods provides a mechanism that prevents the agent from deviating too much from $\pi_b$. In this work, we build on Soft-SPIBB because it has yielded better empirical results. Formally, its constraint on the policy class is defined by:

$$\Pi_{\mathrm{S}} = \left\{ \pi, \text{ s.t. } \forall x, \sum_a e(x,a) \left| \pi(a|x) - \pi_b(a|x) \right| \le \epsilon \right\},$$

where $\epsilon$ is a hyper-parameter that controls the deviation from the baseline policy.

We define $q_{m,\boldsymbol{\lambda}}^\pi(x,a) = \sum_{k \in [d]} \lambda_k q_{m,k}^\pi(x,a)$ to be the state-action value function associated with the linearized $\boldsymbol{\lambda}$ parameters. The same notation extension is used for $v_{m,\boldsymbol{\lambda}}^\pi(x,a)$ and $\mathcal{J}_{m,\boldsymbol{\lambda}}^\pi$. The application of Soft-SPIBB to multi-objective safe policy improvement is therefore direct:

$$\arg\max_{\pi \in \Pi_{\mathrm{A}} \cap \Pi_{\mathrm{S}}} \mathcal{J}_{\hat{m},\boldsymbol{\lambda}}^\pi, \tag{3}$$

which is always realizable since $\pi_b \in \Pi_{\mathrm{A}} \cap \Pi_{\mathrm{S}}$.

We show that the construction of the plausible set required for the application of SPIBB is technically sound by deriving the concentration bounds for the multi-objective case. In Appendix B.1, we show with Hoeffding's inequality that $e$ grows as the square root of the logarithm of $d$ (the number of reward functions), *i.e.* almost imperceptibly. From there, all the SPIBB theoretical results from Laroche et al. (2019); Nadjahi et al. (2019); Simão et al. (2020) may be generalized at a negligible SPI guarantee cost to the multi-objective setting, by applying their theorems separately to every objective function.

Now, the problem in Equation (3) can be transformed into a policy improvement procedure that solves for every state $x \in \mathcal{X}$ the following optimization problem[3]:

$$\pi_{\mathrm{S\text{-}OPT}} = \arg\max_{\pi \in \Pi} \langle \pi(\cdot|x), q_{\hat{m},\boldsymbol{\lambda}}^\pi(x,\cdot) \rangle \tag{S-OPT}$$

$$\text{s.t.} \quad \sum_{a \in \mathcal{A}} e(x,a) \left| \pi(a|x) - \pi_b(a|x) \right| \le \epsilon, \tag{$\pi \in \Pi_{\mathrm{S}}$}$$

$$\forall k \in [d], \sum_{a \in \mathcal{A}} \pi(a|x) \mathop{\mathrm{Adv}}_{\hat{m},k}^{\pi_b}(x,a) \ge 0. \tag{$\pi \in \Pi_{\mathrm{A}}$}$$

The above procedure requires us to make additional algorithmic modifications that are not present in the original SPIBB algorithms. In particular, we need to explicitly incorporate advantage constraints for safety-guarantees for the individual objectives (proof given in Appendix B.2). The classic single-objective SPIBB algorithms do not need to check the advantage conditions because it is automatically guaranteed by the $\arg\max$ and the fact that $\pi_b \in \Pi_{\mathrm{S}}$.

Using the construction above, we directly get the following result on the performance guarantees for each objective function that satisfies the desired property in Equation (1):

**Proposition 3.1.** The policy $\pi$ returned from solving the S-OPT satisfies the following property in every state $x \in \mathcal{X}$ with probability at least $(1 - \delta)$:

$$\forall k \in [d], v_{m^\star,k}^\pi(x) - v_{m^\star,k}^{\pi_b}(x) \ge -\frac{\epsilon v_{\max}}{1 - \gamma}, \tag{4}$$

where $v_{\max} \le \frac{r_\top}{1-\gamma}$ is the maximum of the value function.

The proof is presented in Appendix B.3. The solution of S-OPT is computed by solving the Linear Program using standard solvers, such as cvxpy (Diamond and Boyd, 2016). There is an increase in the computational cost proportional to the number of reward functions. Compared to Soft-SPIBB,

---

[3]In practice, we also need to check $\forall x$ that $\pi(\cdot \mid x)$ is a valid probability distribution: positive and sums to 1.

the value and advantage functions estimation cost increases by a factor of $d$: respectively $\mathcal{O}(d|\mathcal{X}|^3)$ and $\mathcal{O}(d|\mathcal{A}||\mathcal{X}|^2)$. There is $\mathcal{O}(|\mathcal{D}|)$ cost for estimating the error bounds, and we also require solving a Linear Program for each state that approximately amounts to an additional $\mathcal{O}(|\mathcal{X}||\mathcal{A}|^2(|\mathcal{A}| + d))$ steps to the total computational cost (Boyd et al., 2004).

**Remark** (Extension to Constrained-RL). The above methodology can also be extended to the Constrained-RL setting for offline policy improvement in general CMDPs. Recall that SPIBB algorithms offer guarantees in the form of: $v_t - v_b \geq \hat{v}_t - \hat{v}_b - \xi$, where $v_t$ and $v_b$ are respectively the true values of the target and baseline policies, $\hat{v}_t$ and $\hat{v}_b$ are their estimates in the MLE MDP, and $\xi$ is an error term due to parametric uncertainty. As a consequence, any constraint $c$ such that $c \leq v_b + \hat{v}_t - \hat{v}_b - \xi$ may be guaranteed ($v_b - \hat{v}_b$ may easily be bounded with Hoeffding's inequality), and when $c$ is larger, we can return no solution found as with other Seldonian algorithms.

### 3.4 Multi-Objective HCPI (MO-HCPI)

We briefly recall how the HCPI methodology (Thomas et al., 2015a,b) can be applied directly for solving the objective in Equation (2). For a target policy, $\pi_t$, we use $\text{IS}_k(\mathcal{D}, \pi_t, \pi_b)$ to denote the estimated returns for the $k$th reward component ($r_k$) using any IS based off-policy estimator (Precup, 2000). A high-confidence lower bound on $\mathcal{J}_{m^\star,k}^{\pi_t}$ can be defined as:

$$\Pr\Big( \mathcal{J}_{m^\star,k}^{\pi_t} \geq \text{IS}_k(\mathcal{D}, \pi_t, \pi_b) - \text{CI}_k(\mathcal{D}, \delta/d) \Big) \geq 1 - \delta/d, \tag{5}$$

where $\text{CI}_k(\mathcal{D}, \delta) \geq 0$ denotes the terms associated with the choice of concentration inequality employed (and typically $\lim_{|\mathcal{D}| \to \infty} \text{CI}_k(\mathcal{D}, \delta) = 0$).

The dataset $\mathcal{D}$ is first split into train ($\mathcal{D}_{tr}$) and test ($\mathcal{D}_s$) sets by the user. Let $\text{IS}_{\boldsymbol{\lambda}}$ denote the IS estimator associated with the user-specified reward scalarization $\boldsymbol{\lambda}$. Given the user specified parameters: $\boldsymbol{\lambda}, \delta, \text{CI}, \text{IS}, \mathcal{D}_{tr}, \mathcal{D}_s$ and $\pi_b$, the policy improvement problem in Equation (2) is transformed to the following optimization problem:

$$\pi_{\text{H-OPT}} = \arg\max_{\pi \in \Pi} \text{IS}_{\boldsymbol{\lambda}}(\mathcal{D}_{tr}, \pi, \pi_b) \tag{H-OPT}$$
$$\text{s.t.} \quad \forall k \in [d],\ \text{IS}_k(\mathcal{D}_s, \pi, \pi_b) - \text{CI}_k(\mathcal{D}_s, \delta/d) \geq \mu_k,$$

where $\mu_k$ denote the empirical returns for $r_k$ under $\pi_b$. The policy $\pi$ returned by `H-OPT` will only violate the safety guarantees with probability at most $\delta$. Proof of this claim and additional details are provided in Appendix C.

Although we only focus on finite MDPs in this work, the HCPI based approach relies on IS estimates and therefore it can also be used for infinite MDPs or POMDPs. Unfortunately, the IS estimates are typically known to suffer from high variance (Guo et al., 2017). Furthermore, the optimization problem in `H-OPT` is more challenging, and we need to resort to regularization based heuristics.

## 4 Synthetic Experiments

The main benefits of working in a synthetic domain are: (i) we can evaluate the performance on the true MDP instead of relying on off-policy evaluation (OPE) methods, (ii) we have control over the quality of the dataset. We test both MO-SPIBB (`S-OPT`) and MO-HCPI (`H-OPT`) on a variety of parameters: the amount of data, quality of baseline and different user reward scalarizations.

**Env details:** We take a standard CMDP benchmark (Leike et al., 2017; Chow et al., 2018) which consists of a $10 \times 10$ grid. From any state, the agent can move to the adjoining cells in the 4 directions using the 4 actions. The transitions are stochastic, with some probability $\alpha$ (generated randomly for each state-action for every environment instance) the agent is successfully able to reach the next state, and with $(1 - \alpha)$ the agent stays in the current state. The agent starts at the bottom-right corner, and the goal is to reach the opposite corner (top-left). The pits are spawned randomly with some uniform probability ($\eta_{pit} = 0.3$) for each cell. The reward vector consists of two rewards signals. A primary reward $r_0$ that is related the goal and is $+1000.0$ on reaching the goal and $-1.0$ at every other time-step. The secondary reward $r_1$ is related to pits, for which the agent gets $-1.0$ for any action taken in the pit. The constraint threshold for this CMDP is $-2.0$ and $\gamma = 0.99$. Maximum length of an episode is 200 steps. Therefore, the task objective is to reach the goal in the least number of steps, such that the agent does not spend more than 2 time-steps in the pit cells.

**Dataset collection procedure:** For every random CMDP generated, we first find the optimal policy $\pi^*$ by using the procedure described in Appendix D.1. The baseline policy is generated using a convex combination of the optimal policy and a uniform random policy ($\pi_{rand}$), i.e., $\pi_b = \rho\pi^* + (1-\rho)\pi_{rand}$, where $\rho$ controls how close $\pi_b$'s performance is to $\pi^*$. Different datasets with varying sizes and $\rho$ are then collected under $\pi_b$ and given as input to the methods.

**Baselines:** We compare against the following baselines:

- Linearized: This baseline transforms the rewards into a single scalar using $\boldsymbol{\lambda}$ and then applies the traditional policy improvement methods on the linearized objective, i.e, $\arg\max_{\pi\in\Pi} \mathcal{J}^\pi_{\hat{m},\boldsymbol{\lambda}}$.

- Adv-Linearized: This method has the same objective as the Linearized baseline, with the additional constraints based on advantage estimators built from $\hat{m}$, i.e. $\forall x \in \mathcal{X}$:

$$\arg\max_{\pi\in\Pi}\langle\pi(\cdot|x), q^\pi_{\hat{m},\boldsymbol{\lambda}}(x,\cdot)\rangle \tag{6}$$
$$\text{s.t.}\quad \forall k \in [d], \sum_{a\in\mathcal{A}}\pi(a|x)\underset{\hat{m},k}{\overset{\pi_b}{\text{Adv}}}(x,a) \geq 0.$$

**Evaluation:** Using $m^\star$, we can directly calculate the returns for any solution policy. Only tracking the scalarized objective can be misleading, so we track the following metrics:

- Improvement over $\pi_b$: This denotes the difference between the scalarized return of the solution policy and the baseline policy, i.e., $\mathcal{J}^\pi_{m^\star,\boldsymbol{\lambda}} - \mathcal{J}^{\pi_b}_{m^\star,\boldsymbol{\lambda}}$. Mean improvement over $\pi_b$ captures on average improvement over $\pi_b$ in terms of the scalarized objective.

- Failure-rate: The failure rate over $n$ runs captures the number of times, on average, the solution policy ends up violating the safety constraints in Equation (1), and thus performs worse than the baseline. In the context of this task, safety constraints are violated if either the agent takes longer to reach the goal, or it steps into more number of pits compared to $\pi_b$.

We test on different combinations of user preference ($\boldsymbol{\lambda}$) and baseline's quality ($\rho$) on 100 randomly generated CMDPs, where $\lambda_i \in \{0, 1\}$, $\rho \in \{0.1, 0.4, 0.7, 0.9\}$ and $|D| \in \{10, 50, 500, 2000\}$. We evaluate under two settings: (i) we use a fixed set of parameters across different ($\boldsymbol{\lambda}, \rho$) combinations, where we run S-OPT with $\epsilon \in \{0.01, 0.1, 1.0\}$ and H-OPT with Doubly Robust IS estimator (Jiang and Li, 2015) and Student's t-test concentration inequality; (ii) we treat them as hyper-parameters that can be optimized for a particular ($\boldsymbol{\lambda}, \rho$) combination. The best hyper-parameters are tuned in a single environment instance and then they are used to benchmark the results on 100 random CMDPs.

**Results:** The mean results with fixed parameters and $\delta = 0.1$ can be found in Figure 1a. The high failure rate of Linearized baseline, regardless of the size of the dataset, is expected as it optimizes the scalarized reward directly and is agnostic of the individual rewards. Adv-Linearized performs better, but in the low data-regime, we see a high failure rate that eventually decreases as the size of dataset increases. This is expected because with more data, more reliable advantage functions estimates are calculated that are representative of the underlying CMDP. Compared to the baselines, both S-OPT and H-OPT maintain a failure rate below the required confidence parameter $\delta$, regardless of the amount of data. Also, as the size of dataset increases, we see an increase in improvement over $\pi_b$, that makes sense as the methods only deviate from baseline when they are sure of the performance guarantees. We expect S-OPT to violate the constraints with increasing value of $\epsilon$, as it relaxes the constraint on the policy-class (Section 3.3) and leads to a looser guarantee on performance. This again is reflected in our experiments where S-OPT with $\epsilon = 1.0$ has a higher failure-rate than $\epsilon = 0.1$. We observed similar trends for different $\delta$ values. A more detailed plot corresponding to different $\boldsymbol{\lambda}$ and $\rho$ combinations as well as results for a riskier value of $\delta = 0.9$ are given in Appendix D.2.

The results with optimized hyper-parameters can be found in Figure 1b. We notice that when the $\epsilon$ parameter is tuned properly, S-OPT has better performance in terms of improvement over $\pi_b$ for the same amount of samples when compared to H-OPT, while still ensuring the failure rate is less than $\delta$. These observations are consistent with the results in the single-objective setting in the original SPIBB works (Laroche et al., 2019; Nadjahi et al., 2019). The general trends and observations from the fixed-parameter case are also valid here. Additional details, including results for $\boldsymbol{\lambda}, \rho$ combinations, hyper-parameters considered and qualitative analysis can be found in Appendix D.3.

We also compare our methods against Le et al. (2019) in Appendix D.4. We show the advantage of our approach over Le et al. (2019), particularly in the low-data regime, where our methods can

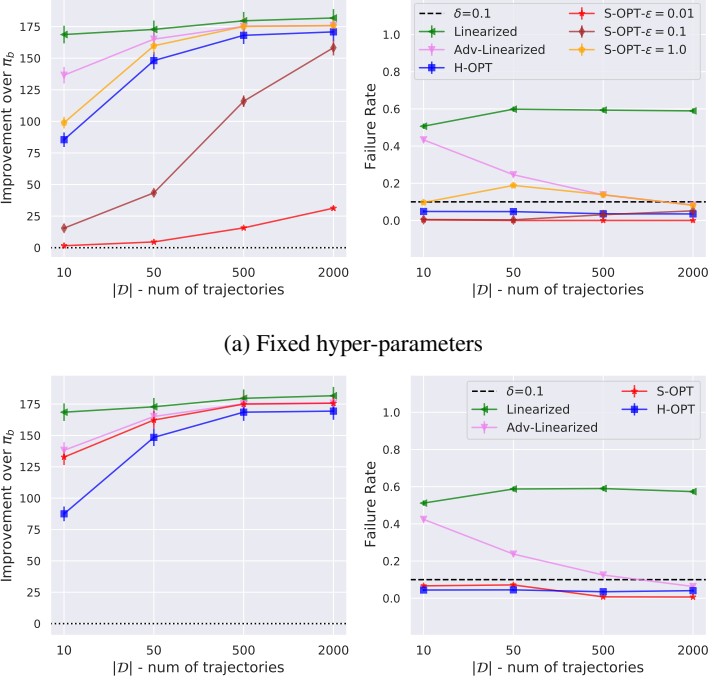

(a) Fixed hyper-parameters

(b) Optimized hyper-parameters.

Figure 1: Results on 100 random CMDPs for different $\boldsymbol{\lambda}$ and $\rho$ combinations with $\delta = 0.1$. The different agents are represented by different markers and colored lines. Each point on the plot denotes the mean (with standard error bars) for 12 different $\boldsymbol{\lambda}$, $\rho$ combinations for the 100 randomly generated CMDPs (1200 datapoints). The x-axis denotes the amount of data the agents were trained on. The y-axis for left subplot in each sub-figure represents the improvement over baseline and the right subplot denotes the failure rate. The dotted black line in the right subplots represents the high-confidence parameter $\delta = 0.1$. Figure 1a denotes when the hyper-parameters are fixed $\epsilon = \{0.01, 0.1, 1.0\}$ and IS = Doubly Robust (DR) estimator with student's t-test concentration inequality. Figure 1b is the version with tuned hyper-parameters for each combination.

improve over the baseline policy while ensuring a low failure rate. This makes sense as the method in Le et al. (2019) relies on the concentrability coefficient which can be arbitrarily high in the low data setting, and therefore their performance guarantees do not hold anymore. We also provide experiments on the scalability of methods with the number of objectives $d$ in Appendix D.5.

## 5 Real-world experiment

In order to validate the applicability of our methods on a real-world task, we consider recent works on sepsis management via RL, where we only have access to a pre-collected patient dataset and goal is to recommend treatment strategies for patients with sepsis in the ICU (Komorowski et al., 2018; Tang et al., 2020). Sepsis is defined as a life-threatening organ dysfunction caused by a dysregulated host response to an infection (Singer et al., 2016). The main treatment method of sepsis involves the repeated administration of intravenous (IV) fluids and vasopressors, but how to manage their appropriate doses at the patient level is still a key clinical challenge (Rhodes et al., 2017).

The problem is safety-critical as our methods need to be cautious about using the data that was possibly collected under unobservable confounders and that can lead to biased model estimates. For instance, a study by Ji et al. (2020) of the model used in Komorowski et al. (2018) found that the learned model suggests clinically implausible behavior in the form of unexpectedly aggressive treatments. We show that our methodology can be applied here to prevent such behavior that results from small sample sizes. We propose to do so by incorporating safety constraints to prevent recommending the treatment decisions that were never or rarely performed in the dataset.

**Data and MDP Construction:** We use the publicly available ICU dataset MIMIC-III (Johnson et al., 2016), with the setup described by Komorowski et al. (2018); Tang et al. (2020) and build on top of their data pre-processing and MDP construction methodology.[4] This leaves us with a cohort of 20,954 unique patients. The state-space consisting of 48 clinical variables summarizing features like demographics, physiological condition, laboratory values, etc., is discretized using a k-means based clustering algorithm to map the states to 750 clusters. The actions include administration of IV fluids and vasopressors, which are categorized into 5 dosage bins each, leading to a total of $|\mathcal{A}| = 25$. The $\gamma$ is set to 0.99. The reward is based on patient mortality. The agent gets a reward, $r_0$, of $\pm 100$ at the end of the episode based on the survival of the patient. More details can be found in Appendix E.1.

In the original work, the rare state-actions taken by the clinicians (state-action pairs observed infrequently in the training set) are removed from the dataset. Instead of removing them, we define an additional reward, $r_1$, based on the rarity of the state-action pair. We define rare state-action pairs to be those that are taken less than 10 times throughout training dataset, and the agent gets a reward of $-10$ for every such rare state-action taken, i.e., $r_1(x, a) = -10.0$ if $\texttt{Count}(x, a) < 10$. The final task objective then becomes to suggest treatments that handles the trade-off between prioritizing improving the survival vs prioritizing commonly used treatment decisions.

**Evaluation:** We compare our approach with the same baselines from Section 4 on different $\boldsymbol{\lambda}$ combinations. We run our methods for 10 runs with different random seeds, where for each run the cohort dataset was split into train/valid/test sets in the ratios of 0.7/0.1/0.2. We evaluate the performance of the solution policies returned by different methods on the test sets using two different OPE methods, Doubly Robust (DR) (Jiang et al., 2015) and Weighted Doubly Robust (WDR) (Thomas and Brunskill, 2016). We acknowledge that these methods are a proxy of the actual performance of the deployed policies. Hence, these results should not be misinterpreted as us claiming that the policies returned by our methods are now ready to be used in the ICU.

Table 1: Performance of various methods using DR and WDR estimators with mean and standard deviation on 10 random splits of the cohort dataset. The red cells denote the corresponding safety constraint violation, i.e, either $\mathcal{J}_0^\pi < \mathcal{J}_0^{\pi_b}$ or $-\mathcal{J}_1^\pi > -\mathcal{J}_1^{\pi_b}$.

| User preference ($\boldsymbol{\lambda}$) | Policy | Survival return ($\mathcal{J}_0$) | | Rare-treatment return ($-\mathcal{J}_1$) | |
|---|---|---|---|---|---|
| | | DR | WDR | DR | WDR |
| | Clinician's ($\pi_b$) | $64.78 \pm 0.90$ | $64.78 \pm 0.90$ | $13.58 \pm 0.19$ | $13.58 \pm 0.19$ |
| $[\lambda_0 = 1, \lambda_1 = 0]$ | Linearized | $97.68 \pm 0.22$ | $97.58 \pm 0.20$ | $27.64 \pm 1.11$ | $27.84 \pm 1.09$ |
| | Adv-Linearized | $91.62 \pm 0.46$ | $92.68 \pm 0.23$ | $15.18 \pm 0.59$ | $13.56 \pm 0.42$ |
| | S-OPT | $66.11 \pm 0.87$ | $66.05 \pm 0.86$ | $13.42 \pm 0.20$ | $13.46 \pm 0.20$ |
| | H-OPT | $65.95 \pm 0.00$ | $65.95 \pm 0.00$ | $13.37 \pm 0.00$ | $13.37 \pm 0.00$ |
| $[\lambda_0 = 1, \lambda_1 = 1]$ | Linearized | $87.17 \pm 0.48$ | $89.11 \pm 0.37$ | $2.41 \pm 0.47$ | $1.52 \pm 0.41$ |
| | Adv-Linearized | $86.77 \pm 0.49$ | $88.58 \pm 0.25$ | $2.53 \pm 0.50$ | $1.57 \pm 0.43$ |
| | S-OPT | $86.77 \pm 0.49$ | $88.58 \pm 0.25$ | $2.53 \pm 0.50$ | $1.57 \pm 0.43$ |
| | H-OPT | $86.37 \pm 0.00$ | $88.03 \pm 0.00$ | $2.58 \pm 0.00$ | $1.43 \pm 0.00$ |
| $[\lambda_0 = 0, \lambda_1 = 0]$ | Linearized | $-89.39 \pm 0.43$ | $-90.90 \pm 0.29$ | $22.99 \pm 0.40$ | $22.81 \pm 0.30$ |
| | Adv-Linearized | $60.27 \pm 0.49$ | $61.44 \pm 0.85$ | $18.40 \pm 0.27$ | $15.36 \pm 0.58$ |
| | S-OPT | $67.73 \pm 0.82$ | $67.22 \pm 0.88$ | $13.24 \pm 0.24$ | $13.55 \pm 0.33$ |
| | H-OPT | $65.95 \pm 0.00$ | $65.95 \pm 0.00$ | $13.37 \pm 0.00$ | $13.37 \pm 0.00$ |
| $[\lambda_0 = 0, \lambda_1 = 1]$ | Linearized | $58.27 \pm 2.18$ | $60.52 \pm 2.07$ | $0.04 \pm 0.03$ | $0.02 \pm 0.01$ |
| | Adv-Linearized | $76.05 \pm 0.65$ | $76.85 \pm 0.72$ | $0.07 \pm 0.05$ | $0.04 \pm 0.03$ |
| | S-OPT | $76.07 \pm 0.65$ | $76.87 \pm 0.73$ | $0.07 \pm 0.05$ | $0.04 \pm 0.03$ |
| | H-OPT | $76.54 \pm 0.00$ | $77.55 \pm 0.00$ | $0.09 \pm 0.00$ | $0.05 \pm 0.00$ |

**Results:** We refer to the return associated with the mortality reward ($r_0$) as survival return ($\mathcal{J}_0$), and the negative return associated with rare state-action reward ($r_1$) as rare-treatment return ($-\mathcal{J}_1$). Higher survival return implies more successful discharges, and lower rare-treatment return implies more adherence to common practice treatment decisions. We present the results on survival and rare-treatment returns in Table 1. As expected, we observe both the Linearized and Adv-Linearized baselines violates constraints across different $\boldsymbol{\lambda}$, whereas S-OPT and H-OPT are able to respect the

---

[4]A caveat here is regarding the underlying assumption that the MDP construction methodology by Komorowski et al. (2018); Tang et al. (2020) maintains the Markovian property in the discretized state-space.

safety constraints irrespective of the $\boldsymbol{\lambda}$.[5] The validation set was used to tune the hyper-parameters, and we report how the performance varies with different hyper-parameters in Appendix E.2.

**Qualitative Analysis:** We conclude with a qualitative analysis of the policies returned from our setting and the traditional RL approach of maximizing just the survival return. Ji et al. (2020) found that the RL-policies for sepsis-management task usually end up recommending aggressive treatments, particularly high vasopressor doses for states where the common practice (according to most frequent action chosen by the clinician for that state) is to give no vasopressors at all. The common practice involves giving zero vasopressors for 722 of the 750 states. However, the policy returned by the traditional single-objective RL baseline recommends vasopressors in 562 (77.84%) of those 722 states, with 295 of those recommendations being large doses (upper 50th percentile of nonzero amounts or $> 0.2$ $\mu$g/kg/min). We compare these statistics for two of the policies returned by MO-SPIBB that deviate the most from $\pi_b$. The policy returned by S-OPT ($\boldsymbol{\lambda} = [1, 1]$) recommends vasopressors in only 93 of those states (12.88 %), with 47 of those recommendations belonging to high dosages. The other policy, S-OPT ($\boldsymbol{\lambda} = [0, 1]$), recommends vasopressors in 134 (18.56 %) of those states and 70 of those recommendations fall in large dosages. Therefore, the policies returned by our approach, even when they deviate from the baseline, are less aggressive in recommending rare treatments. In Appendix E.3, we present an additional qualitative analysis that demonstrates our methods recommend lesser rare-action treatments than the traditional single-objective RL approach.

An argument can be made against the case when all rare state-action pairs are removed from the training data itself. This will ensure that any learned policy will have near 0 rare-treatment return. However, it is not always clear how to define the cut-off criteria for rare-actions, and it might be possible that some of these rare state-action pairs are actually crucial for finding a better policy. For instance, we did an experiment where we assigned state-actions pairs with frequency $< 100$ to be rare state-action pairs and filtered those from the training set. The clinician's performance on the test set using a DR estimator for survival return is 65.95. In this case, the traditional single-objective RL baseline gives the survival return of 11.26, which shows that removing such transitions from the dataset actually hampers the solution quality. Our approach of assigning a separate reward for rare state-action pairs is able to find a solution with a survival return of 86.75 even in this scenario.

## 6  Conclusion

We present a new Seldonian RL algorithm that takes the user preference based scalarization into account while ensuring the solution policy performs reliably in context to the baseline policy across all objectives. On both synthetic and real-world tasks, we show that the proposed approach can improve the policy while ensuring the safety constraints are respected.

Our setting can accommodate any general form of scalarizations (e.g. non-linear or convex) as well as objectives (such as fairness), making it applicable to a wide variety of real-world tasks. The only assumption we made is regarding the dataset being collected under a single known baseline policy. An exciting line of future work can be to relax this assumption and consider the scenario where the dataset comes from a variety of unknown policies with different qualities. We did not make any claims about the optimality of the solutions as often optimality and safety are contradicting objectives. It is not clear how (and if) one can make claims about optimality in the offline setting without bringing in additional unrealistic assumptions (Section 2). The extension to infinite MDPs and the function-approximation setting is also left for future work. It is important to note that when it comes to practical application, it is not unusual for continuous domains to be discretized to enable better interpretability, especially when interactions with humans are necessary. If the Markovian property is valid in the discretized space, SPIBB-based guarantees will also hold true.

## 7  Acknowledgements

The authors would like to thank NSERC (Natural Sciences and Engineering Research Council), IVADO (Institut de valorisation des données) and CIFAR (Canadian Institute for Advanced Research) for funding to McGill in support of this research. Philip S. Thomas was funded in part by NSF award

---

[5]In Table 1, $\boldsymbol{\lambda} = [1, 1]$ represents a rare case of reward scalarization that allows all the methods to find a good solution policy that satisfies the constraints. In general, it is difficult to find such scalarization parameters as seen in synthetic experiments (Appendix D.2).

#2018372. The computational component of this research was enabled in part by support provided by Calcul Québec (`www.calculquebec.ca`), Compute Canada (`www.computecanada.ca`) and Mila's IDT team.

We would also like to thank Emmanuel Bengio and Koustuv Sinha for many helpful discussions about the work, and the anonymous reviewers for providing constructive feedback.

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
