# OpenReview forum: "Multi-Objective SPIBB: Seldonian Offline Policy Improvement with Safety Constraints in Finite MDPs"
_NeurIPS.cc/2021/Conference — NeurIPS 2021 Poster_

### Official Review · Reviewer_VZ8L · 2021-07-15

**Rating:** 6
**Confidence:** 4

**Summary:**

The authors study the safe policy improvement problem in the offline RL settings where the known baseline policy is available. In this problem settings, the dataset has been already collected by the baseline policy and contains multiple reward signals from the environment. Under this problem setting, the authors aim to achieve a policy that balances the different rewards while guaranteeing that the new policy is better than the baseline policy. The authors proposed a new algorithm based on SPIBB and show its effectiveness in a synthetic environment and the one with sepsis management.

**Limitations And Societal Impact:**

limitations and societal impact have been fully addressed.

**Main Review:**

I think this paper is well-written. The problem considered in this paper is reasonable and important for the research community. Though the assumption on the baseline policy (i.e., Assumption 3.1) is rather strong, there are indeed scenarios where we can assume the availability of the known baseline policy.

Pros
- Well-written texts. Though I think there is space for improvement as noted in Minor Comments, this paper is mostly easy to understand. Motivations are clear and algorithms are well-presented.
- The numerical experiments are well-designed and the results are promising (though I have several criticisms as noted in Cons).
- Reproducibility. I read through the source code (did not run, though), and I found it easy to read. This source would be helpful for future research.

Cons
- Incremental theoretical analysis. Theoretical results are simple extensions of the previous work. I don't think there are strong contributions in the theory parts.
- I do feel that the authors should have (also) tested their algorithm in the standard benchmark. (e.g., MuJoCo tasks) as in Urpi et al. (2020).
    - Urpí, Núria Armengol, Sebastian Curi, and Andreas Krause. "Risk-Averse Offline Reinforcement Learning." ICLR. 2020.
- Lack of comparison with previous work. The authors compared their algorithms with very simple baselines (i.e., minor modification with theirs). For example, the authors should have compared their method with such methods as in Le et al. (2019).
    - Le, H., Voloshin, C., and Yue, Y. (2019). Batch policy learning under constraints. In ICML.

### Minor Comments:
- I think this paper would be easier to read if the authors use the same characters as the one typically used in the RL paper. The original SPIBB paper (https://arxiv.org/pdf/1712.06924.pdf) is much easier to read, at least for me. For example, I would like the authors to change as follows:
    - $q \rightarrow Q$
    - $v \rightarrow V$
    - $r_{\top} \rightarrow R_{\max}$

- The authors may consider including Reward-free RL papers in the related work. As a notable one,
    - Jin, Chi, et al. "Reward-free exploration for reinforcement learning." In ICML. 2020.

### Questions
- The descriptions in lines 74 - 81. Could you elaborate more? Why can you say that the assumptions in this paper are milder than those in the previous work? And, could you provide the reason why the authors did not have to make the concentrability assumption in this paper?


---
Score is updated according to the new experimental result.

**Time Spent Reviewing:**

5 hours

---

> ### Author Response · Authors · 2021-08-10
> **Response to Reviewer VZ8L**
>
> Thank you for your helpful review and criticisms. We are glad you found the problem we address in this work to be relevant, and the methodology and results of the accompanying experiment to be comprehensive. We address your concerns below:
>
> * **Incremental theoretical analysis:**
>     While we agree that the theoretical analysis is rather straightforward, the extension of SPIBB to the multi-objective setting also requires us to make additional algorithmic modifications that require incorporating advantage constraints for individual objectives in the optimization problem (Appendix B.2). Even though this formulation is a natural extension, it is technically sound and provides a non-trivial advantage, and can achieve our requirements regarding practical safety guarantees.
>
> * **Experiments on Mujoco tasks:**
>     The main motivation of our work is to do policy improvement in a safety-critical setting where we can have practical high-confidence guarantees on the performance of the policies returned by our algorithm. Note that this is presently not the case for the Deep-RL approaches that are applied to the Mujoco tasks. For instance, the work in [1] does not provide any high-confidence guarantees on the performance of the solution policy, either theoretically or empirically.  Hence, we focus our approach in the finite MDPs setting where it is theoretically and empirically grounded.
>
>     It’s important to note that when it comes to practical application, it is not unusual for continuous domains to be discretized to enable better interpretability, especially when interactions with humans are necessary. For instance, in the Sepsis task, the domain space is high-dimensional where the patient data consists of a 48-dimensional time series with features representing attributes such as demographics, vitals and lab work results (Table 3 in Appendix). A non-parametric function approximation scheme (k-means based clustering) is used to discretize the underlying space.  If the Markovian property is valid in the discretized space, the SPIBB-based guarantees will also hold. Thus the finite case is in fact very useful in practice.
>
>
>
> * **Comparison with Le et al [2]:**
> As we mention briefly in the related work, the performance guarantees in [2] rely on the concentrability assumption, an assumption that cannot be verified in practice. Concentrability is a strong assumption that upper bounds the ratio between the future state-action distributions of any non-stationary policy and the baseline policy under which the dataset was generated by some constant. From a practical perspective, it is unclear how to get a tractable estimate of this constant, as the space of future state-action distributions of non-stationary policies is vast and as such this assumption can not be verified in practice. Thus, this constant can be arbitrarily huge, potentially even infinite when the baseline policy fails to cover the support of the space of all non-stationary policies (such as in the low-data regime we look at in our experiments), leading to the performance bounds given by these methods to blow up (and even be unbounded). Additionally, these guarantees are only valid with respect to the performance of the optimal policy. From a safety-critical viewpoint, this goes against our motivation as we want practical guarantees on the performance of the policies that we want to eventually deploy.  Therefore, it becomes very complex to design experimental scenarios where we can extend the methodology by [2] and ensure a fair comparison.
>
> * **Questions regarding concentrability (lines 74-81):**
>     Please refer to the point above on comparison with [2] for more information on concentrability assumption and why it is not a viable option from a safety-critical application perspective. As we do not rely on the concentrability assumption, we are not making any assumptions regarding the distribution of state-action pairs visited by the baseline policy. Moreover, we are not making any assumptions or claims with respect to the optimal policy. The only assumption we make is that we have access to the baseline policy under which the dataset was collected (or that baseline policy can be estimated from the dataset itself). As we do not need to make any assumptions regarding the distribution of the transitions in the dataset, we can argue that the assumptions we make are milder than the work in [2].
>
>      The concentrability assumption is typically used to bound the error of the Bellman optimality operator of the Q-learning update that might occur due to sampling and function approximation. The reason we do not need to make concentrability assumptions is that: (1) we are not making any guarantees with respect to the performance of the optimal policy, and (2) we are in the finite MDPs setting.
>
>
> We failed to make the above points clear in the draft, and we hope that our responses above provide the right perspective on the work. Thank you for pointing out this possible point of confusion. We will update the draft to paper to clarify these points. Please let us know if we have addressed your concerns adequately or otherwise if there are any remaining concerns.
>
>
> References:
> [1] Urpí, Núria Armengol, Sebastian Curi, and Andreas Krause. "Risk-Averse Offline Reinforcement Learning." ICLR. 2020.
> [2] Le, H., Voloshin, C., and Yue, Y. (2019). Batch policy learning under constraints. In ICML.

---

> > ### Comment · Reviewer_VZ8L · 2021-08-25
> > **Thank you for response**
> >
> > Thank you for your clarifications.
> >
> > **Experiment.** I intended to ask the reasons why standard RL benchmark environments were not used (MuJoCo was just an example). If the authors use a more commonly used benchmark, future comparison will become much easier for other researchers. Anyway, I understand why the authors chose the environments in this paper.
> >
> > **Concentrability.** Thank you for answering my question.
> >
> > **Comparison w/ Le et al. (2019).** I read the discussion between the authors and Reviewer q8Um. If new experimental results are provided, I will consider whether I should change my score or not.

---

> > > ### Author Response · Authors · 2021-08-28
> > > **Empirical comparisons provided for Le et al. (2019) in new thread**
> > >
> > > Thank you for waiting. We have provided the empirical comparisons with Le et al. in a new discussion thread titled “Le et al. (2019) Comparison Results” addressed to all reviewers.
> > >
> > > Please let us know if you have any questions or concerns that you would like us to respond to or require any additional information. Thanks for your time.

---

> > > > ### Comment · Reviewer_VZ8L · 2021-08-31
> > > > **Thank you for additional results**
> > > >
> > > > Thank you for providing new results. I think the authors' claims are now better supported than before. Reading the authors' comments, the new experimental results seem convincing and well-conducted.
> > > >
> > > > Though this is an issue of NeurIPS review process (say, revised versions cannot be uploaded in OpenReview), I cannot strongly recommend acceptance before actually reading the paper. I updated my score slightly more positive one (5 --> 6) according to the new experimental results.
> > > >
> > > > Leaving aside whether this paper will be accepted or not at this conference, I think the quality of the paper has improved by the new experimental results. I would like to appreciate the authors' effort.

---

> > > > > ### Author Response · Authors · 2021-08-31
> > > > > **Thank you!**
> > > > >
> > > > > Thank you for engaging with our response, for your helpful suggestions, and for revising your score. It's much appreciated.

---

### Official Review · Reviewer_VHxZ · 2021-07-15

**Rating:** 7
**Confidence:** 4

**Summary:**

The paper proposes a new Safe Policy Improvement problem formulation to deal with multi-objective offline settings.

Authors consider a problem where a batch of data, collected by following a known baseline policy, is given and the goal is to return a new policy that, achieves (with high confidence) better results than the baseline, considering also user’s preferences.

Taking inspiration from the Seldonian approach [Thomas et al., 2019] and offline SPI algorithms (SPIBB [Laroche et al., 2019], Soft-SPIBB [Nadjahi et al., 2019], HCPI [Thomas et al., 2015a,b]), the authors then show that this can be formulated as particular optimization problems which can be solved, by using common techniques with a computational penalty that scales with the number of objectives, to obtain a policy with the required safety guarantees.

Finally, they compare their methods (MO-SPIBB and MO-HCPI) against two baselines on two experimental settings, a synthetic problem and real-world scenario, showing that their approaches are, when optimized, very competitive but also safe, obtaining results that are consistently better than the baseline up to the accuracy desired, especially in a small sample regime.


**Limitations And Societal Impact:**

The limitations and societal impacts are clearly stated by the authors, in the main paper and in the checklist.


**Main Review:**

Strengths

The paper is well written: it is clear what the authors want to achieve and the context of the work.

The paper addresses a relevant problem, safe algorithms in RL, in a setting which has not been extensively considered before (multi-objective).

The authors also present a well-rounded experimental section: a synthetic environment and an experiment on real-world data. The latter is especially interesting because it gives an idea on how the methods can be actually be applied to real-life problems.

Weaknesses

The work is extensively based on existing approaches and its contribution (in term of novelty ) is not major: the proposed methods amount to solve a new optimization problem and lead to solutions that can only be applied to fully-observable, discrete multi-objective environments, as noted by the authors; considering the paper’s focus on real-world applications, this is a limiting factor.

Also, the experimental settings are on the small side and thus do not give a clear idea on how the methods would behave on more realistic models, especially in terms of their performance.

Clarity
The paper is well written. In particular, it is clear what problem the authors are addressing, the properties that they want their method to have, what they want to achieve and the paper’s contributions.

The experimental methodology and results are also well explained.

However, there are some small notation issues or confusing details that I think should be considered by the authors:

1) Line 115: the multi-reward signal is defined to be stochastic but is given as a vector-valued function from state-action pairs to a d-cube, i.e. mapping pairs to d scalars. Shouldn’t be to a distribution over scalars?

2) Same line: $r_{\top}$ is not defined, although not hard to infer

3) Equation (S-OPT), after line 186: as in 2), I would specify that the brackets $\langle \rangle$ are to be read as an inner product. Moreover, it would be better to separate the two operands by a comma or a vertical line, or use the dot notation instead

4) Same equation: it is not immediately clear from the main paper why the third line of this equation is something necessary to guarantee the condition of belonging to $\Pi_A$. A brief explanation would significantly help here.

5) Section 4, Dataset collection procedure: do you collect only one dataset for dataset size for each one of the 100 random CMDPs and twelve $\rho$?

6) Why do some lambdas parameters sum to 2 or 0 in Section 5, Table 1? In line 141 lambda is defined over the d-simplex. Also, for the case $\lambda_0 = \lambda_1 = 0$, does this mean that the [action-]value function is zero for every state[-action pair] (line 174)?

Originality

The methods extend known SPI algorithms to deal with a new problem and, although not completely original, they differ from existing techniques in the sense that they amount to solve new optimization problems, with an added constraint; also, it is clear from the paper how these new methods differ from previous work, and how it is situated with respect to related contributions.

Significance

This paper presents an extension of common SPI approaches that can handle multi-objective environments, so its contribution could be definitely very useful to researchers, perhaps as a stepping stone for other methods with slightly different assumptions or approximation techniques.
However, the fact that this paper doesn’t consider relevant real-life modeling problems, such as partial observability or function approximation, somewhat limits its significance, especially for practitioners.

Quality

The claims are supported, even though the mathematical proofs are completely relegated to the supplementary material; the experimental methodology is also technically sound, and extensively explained. The authors are also clear about the limitations of their contribution, its strengths and weaknesses, although I would have liked some comments about the computational limits of these approaches, and how they would fare in very large experiments.


**Time Spent Reviewing:**

10

---

> ### Author Response · Authors · 2021-08-10
> **Response to Reviewer VHxZ**
>
> Thank you for taking the time to read our work and writing an encouraging and very thoughtful review of the paper.
>
> 1. *"Notation issues in Line 115, Line 141"*
> You are absolutely correct in pointing out the notational inconsistencies. Thank you for bringing this to our attention. We will update the draft to address these errors.
>
> 2. *"Section 4, dataset collection procedure"*
> For each random CMDP, and each baseline policy ($\rho$ value), we generate a new dataset according to the dataset size. The accompanying method can be found at `gridworld/core/run_PI_agent.py` at Line 135-150.
>
> 3. *"comments on computational limits of these approaches"*
> Our methods requires estimating the value and advantage functions that can be calculated in $\mathcal{O}(d|\mathcal{X}|^3)$ and  $\mathcal{O}(d|\mathcal{A}||\mathcal{X}|^2)$  steps respectively. MO-SPIBB requires $\mathcal{O}(|\mathcal{D}|)$ for estimating the error bounds, and it also requires solving a Linear Program for each state that approximately amounts to an additional $\mathcal{O}(|\mathcal{X}||\mathcal{A}|^2(|\mathcal{A}|+d))$ steps to the total computational cost [1]. Similarly, for MO-HCPI, there is an increase in the cost of the Importance Sampling (IS) estimation step by a factor of $d$ (where the cost of IS step depends on the choice of IS based off-policy estimator) along with an increase in the cost of applying concentration inequalities step by a factor of $d$, i.e., $\mathcal{O}(d|\mathcal{D}|)$.
>
>     Although we did not test the computational limits w.r.t. the number of states and actions, we did a scaling study with respect to the number of objectives in the grid-world CMDP domain (Appendix D.4).
>
>
> References:
>
> [1] Boyd, Stephen, Stephen P. Boyd, and Lieven Vandenberghe. Convex optimization. Cambridge university press, 2004. (Section 1.2.2, Linear programming)

---

> > ### Author Response · Authors · 2021-08-31
> > **Any further clarifications required?**
> >
> > Dear Reviewer,
> >
> > Thank you again for your valuable feedback and comments that will help us improve this work.
> >
> > Since the discussion phase of the review period is closing soon, please let us know if everything about the submission is clear now.
> >
> > Thanks for your time and consideration. It is much appreciated.

---

### Official Review · Reviewer_q8Um · 2021-07-16

**Rating:** 6
**Confidence:** 3

**Summary:**

The paper describes the multi-objective extension of the SPIBB (Laroche et al, 2019) and Soft-SPIBB (Nadjahi et al., 2019) methods for policy improvement with respect to a number of constraints. It is performed by incorporative multi-objective state-action value functions into the original Soft-SPIBB method.


**Ethical Concerns:**

All concerns acknowledged and addressed by the authors

**Limitations And Societal Impact:**

The authors correctly addressed limitations such as only considering finite Markov Decision Processes (MDPs); the authors addressed in their comment potential negative impact of their work such as misuse of reinforcement learning.

**Main Review:**

*Pros: *

Interesting and important problem to address (see below)

*Cons:*

 Insufficient experimental analysis, novelty is not clear as the narrative describes SPIBB extension for multi objective training by incorporating linear combination of state-action value functions; the clarity and reproducibility could be improved .

While the reviewer thinks that multi-objective reinforcement learning with safety constraints is a very important problem, the reviewer also wonders whether evaluating improvements of the policy over one user-chosen preference (as opposed to building a Pareto front) is sufficient and is enough for solving a multiobjective problem. If we only find some of the solutions for some pre-defined constraint, it may not explain the trade-offs between multiple objectives, which, as the reviewer understands, is a motivation behind multi-objective optimisation. An introduction in a popular book chapter in multiobjective optimisation [1] explains this motivation as follows:

*'Many real-world search and optimization problems are naturally posed as non-linear programming problems having multiple objectives. Due to the lack of suitable solution techniques, such problems were artificially converted into a single-objective problem and solved. The difficulty arose because such problems give rise to a set of trade-off optimal solutions (known as Pareto-optimal solutions), instead of a single optimum solution. It then becomes important to find not just one Pareto-optimal solution, but as many of them as possible. This is because any two such solutions constitutes a trade-off among the objectives and users would be in a better position to make a choice when many such trade-off solutions are unveiled.'*

*Reproducibility*

It is not clear how exactly the baseline policy was generated: what were the hyperparameters of the method described in D1, what optimisation method was used and how was the policy function defined (tabular?)? Line 198 in the description of the proposed method states: "The solution of S-OPT is computed by solving the Linear Program using standard solvers, such as cvxpy (Diamond and Boyd, 2016)." Was the same method used for generating the policy, what exact solvers  have been used, and what were the hyperparameters of the solvers? It would be good to state it in the appendix and reference it from the experimental section.


*Other comments:*

- The reviewer thinks that it would be good to structure the paper in a way that outlines novel contribution to answer the following: apart from changing the optimisation to multiobjective, what would differentiate this work from Soft-SPIBB? The reviewer finds it hard to figure it out from 3.3 and 3.4.  One approach could be to define it as an algorithm and outline the new parts of it.

- Line 216: “The dataset D is first split into train (Dtr ) and test (Ds) sets by the user.” How exactly does it happen and why is it not possible to use randomisation instead?

- In Eq. 1 we have the constraint expressed via parameters $\delta$ and $\zeta$; from line 172 onwards, we see a different set of constraints without parameter $\zeta$. Could the authors explain the transition between eq. 1 and the problem (S-OPT)?

==
The score updated to 6 taking into account the rebuttal (see follow up discussion). l the confidence decreased from 4 to 3 due to large number of changes discussed during the rebuttal.

[1] Deb K. (2005) Multi-Objective Optimization. In: Burke E.K., Kendall G. (eds) Search Methodologies. Springer, Boston, MA. https://doi.org/10.1007/0-387-28356-0_10

**Time Spent Reviewing:**

6

---

> ### Author Response · Authors · 2021-08-10
> **Response to Reviewer q8Um**
>
> Thank you for your feedback and criticisms. We understand you have main concerns about (1) the setting and choice of user preference as an input, (2) more details regarding experimentation methodology, and (3) differentiation with SPIBB. We have tried to address the points you raised below:
>
> **Optimal Pareto-frontier vs user-preference as input:**
> You are correct in stating that one way to evaluate the Multi-objective problems is via the *multiple-policy* [1,2] MORL approaches that compute the policies that approximate the true optimal Prato-frontier. However, note that optimality and safety are contradicting objectives. It is not clear how (and if) one can make claims about optimality in the offline setting without bringing in additional unrealistic assumptions (Section 2, Related work on MORL). Instead, we take an alternate approach inspired by another category of Multi-Objective problems called *single-policy* [2,3] approaches where the trade-offs between different objectives are controlled by the user via providing a scalarization or user-preference vector ($\boldsymbol{\lambda}$) as input to the algorithm.  The goal then becomes to maximize the objective specified by the user via $\boldsymbol{\lambda}$. However, the users might make mistakes in specifying this objective, and we offer guarantees that prevent deteriorating the performance of the policy across any objective.
>
> To clarify, take the example of the study in the Sepsis task in Section 5 when there are two reward functions: one that is related to patient survival, and another that is related to the confidence of the physician in the treatment strategies (such as rare or aggressive treatment decisions). In this setting, the safety constraint that we have is: do not change the policy unless you can guarantee the survival return will improve without increasing the risk of the treatment decisions. Our goal here is *not* to reason about a frontier but ensure that the expected return w.r.t. this reward function is at least some threshold. We show in our qualitative analysis, that our methods can achieve this goal and are able to improve the survival return while being less aggressive in recommending riskier treatments compared to the other approaches across different $\boldsymbol{\lambda}$ values. The trade-off between prioritizing improving the survival vs prioritizing including commonly used low-risk treatment decisions is explicitly controlled by the user via specifying the $\boldsymbol{\lambda}$ parameter.
>
> Finally, one could imagine an iterative process with the user refining its scalarization in the function of the expected trade-offs returned by the algorithm. Similar approaches are also taken in multiple-policy MORL literature where a single-policy algorithm is called repeatedly with different scalarizations [2,4]. We argue that our work solves this important problem by allowing the user to experiment with different reward design strategies in safety-critical settings without worrying about the risks of ill-defined scalarizations.
>
>
>
> **Reproducibility:**
> * Hyper-parameters for calculating the optimal policy of a CMDP (Appendix D.1): There is *NO* hyper-parameter associated with this method. Note that the method in Appendix D.1 requires a model of the CMDP as an input to calculate the corresponding optimal policy exactly via Linear Programming. The optimization variables are the occupancy measure, i.e., the future state-action distribution $\rho^{\pi}(x,a)$. We used CVXPY for solving this LP with the default parameters. From the documentation of CVXPY [5], "by default CVXPY calls the solver most specialized to the problem type", and as such we don't treat the optimization method as a hyperparameter. The corresponding method is implemented as `cmdp_dual_lp`  in the file `gridworld/core/utils.py` (Lines 95-144).
>
> * Solution for S-OPT: The method for solving S-OPT is implemented in the `make_policy_iteration_operator` in the `ConstSPIBBAgent` class in the file `gridworld/agents/tabular_spibb_agents.py`. As with the previous case, we used the default solver choice by CVXPY and didn't treat the optimization solver as a hyper-parameter. The same termination condition value for generating the iterates of the stochastic approximation methods, along with the same value of the maximum number of iterations, were used across different methods in different settings. These values are provided in the corresponding launch scripts, for instance in `gridworld/scripts/large_grid_exp.py`. We thought these details might make the Appendix cumbersome, so we left them in the accompanying code in an effort to reduce redundancy.
>
> Please note that all the accompanying code used for running the experiments and generating the plots is provided in the Supplementary Materials. In this regard, the other reviewers have also acknowledged our efforts regarding the reproducibility and thoroughness of the experiment methodology.
>
> Finally, you mention in the review that there is insufficient experimental analysis, but it is unclear what part do you refer to while making that statement. Please let us know where your concerns regarding experimental analysis lie, so we have the opportunity to address them.
>
> **Differentiation with SPIBB:**
> Along with changing the optimization to a multi-objective case and ensuring that the construction of the plausible set required for the application of SPIBB is valid in this new setting, we explicitly incorporate advantage constraints for safety guarantees across the individual objectives (Appendix B.2). This adds additional constraints in the optimization problem that are not present in the original Soft-SPIBB formulation. Additionally, the experimental methodology, including the evaluation of the grid world CMDPs and the application of MO-SPIBB and MO-HCPI to the Sepsis study, have not been considered in the earlier work and are an important empirical contribution that will be also useful to any future work in this domain.
>
>
> **Other comments:**
> * Line 216, dataset split for HCPI: The split of the dataset into train and test is an essential component of the HCPI methodology where the user needs to define the ratio of the split. In line with the previous work, we used a split of 0.7:0.3 for the train and test sets. This param can be found in `HCPIAgent` class in the file `gridworld/agents/tabular_hcpi_agents.py` at line 347 (`training_size = 0.7,`).
>
> * Constraints via $\delta$ and $\xi$: In S-OPT, the $\delta$ is included in the error function $e(x,a)$ and the precision parameter $\xi$ is now replaced with the hyper-parameter $\epsilon$. This was done to be consistent with the earlier work on SPIBB and Soft-SPIBB.
>
>
>
> We hope we have sufficiently addressed your concerns and reassured you that the problem we tackle is important and our methodology is technically and empirically sound. We hope this clears any misunderstandings and you see our work in a favourable light. Please let us know if you're not convinced by our arguments or if you have any additional questions.
>
>
> References:
>
> [1] Vamplew, P., Dazeley, R., Berry, A., Issabekov, R., and Dekker, E. Empirical evaluation methods for multiobjective reinforcement learning algorithms. Machine Learning, 84(1):51–80, Jul 2011.
>
> [2] Roijers, D. M., Vamplew, P., Whiteson, S., and Dazeley, R. (2013). A survey of multi-objective sequential decision-making. Journal of Artificial Intelligence Research, 48:67–113.
>
> [3] Van Moffaert, K., Drugan, M. M., and Nowe ́, A. Scalarized multi-objective reinforcement learning: Novel design techniques. In Proceedings of the IEEE Symposium on Adaptive Dynamic Programming and Reinforcement Learning (ADPRL), pp. 191–199, 2013.
>
> [4] Zuluaga, M., Krause, A., and Pu ̈schel, M. ε-pal: An active learning approach to the multi-objective optimization problem. Journal of Machine Learning Research, 17(1): 3619–3650, 2016.
>
> [5] https://www.cvxpy.org/tutorial/advanced/index.html?highlight=solvers

---

> > ### Comment · Reviewer_q8Um · 2021-08-24
> > **Definitely improves the standing of the paper, however, there is one outstanding question**
> >
> > First of all, I would like to thank the authors for the clarity of answers.
> >
> > On clarity and reproducibility: the answers sound good to me.
> >
> > On motivation and  setting: the answer is thorough and gives a good argument, and I am looking forward to seeing it in the updated paper.
> >
> > Based on that I increase the score.
> >
> > After reading other reviews, and following up on my own question on experimental analysis, I see that the main outstanding concern is about experimental backup of the proposed methods (the other reviewers recommended MujoCo tasks and Le et al (2019), but I understand from the authors' comments that it is not possible). It is still not clear if the authors propose any way to address it. Is there any plan to show the results with stronger baselines or more real-world dataset tasks additional to sepsis dataset? Alternatively, what is the authors' strategy to improve on the experimental section?

---

> > > ### Author Response · Authors · 2021-08-24
> > > **Thanks for the score revision, extension of Le et al. (2019) to our setting is in progress**
> > >
> > > Thank you for the response and the score revision. We are glad we were able to alleviate your concerns regarding the motivation, setting and reproducibility.
> > >
> > > Thanks again for the suggestions on making the experimental section stronger. We are working on an implementation of Le et al. (2019) extended to our setting that we will include as a baseline.
> > >
> > > Currently, we have the experiments on a synthetic environment (grid-world CMDP) and complex real-world data (Sepsis study). Please let us know if there are any particular multi-objective safety-critical RL environments that we should consider to improve the experimental section.

---

> > > > ### Comment · Reviewer_q8Um · 2021-08-25
> > > > **Implementation of Le et al (2019) sounds good**
> > > >
> > > > Thank you for the prompt answer. Implementation of Le et al (2019) sounds good and would help improve the standing of experimental evaluation. The remaining question is as follows: to  make the score change towards acceptance, I would need to see more material details on how you are improving it. It means: do the authors have any experimental results from Le at al (2019) to share?

---

> > > > > ### Author Response · Authors · 2021-08-28
> > > > > **Comparisons provided for Le et al. (2019) in new thread**
> > > > >
> > > > > Thanks for your patience. We have provided the comparisons with Le et al. in a new discussion thread titled "Le et al. (2019) Comparison Results" addressed to all reviewers.  We also answer your question regarding experimental results and environments from Le et al. in the same comment in the "Environments in Le et al." section.
> > > > >
> > > > > Please let us know if you have any questions or concerns that you would like us to respond to or require any additional information. Thanks for your time.

---

> > > > > > ### Comment · Reviewer_q8Um · 2021-08-30
> > > > > > **I see that there is an improvement over Le et al's failure rate**
> > > > > >
> > > > > > Thank you for providing a thorough response and a new set of experiments, it improves the standing of the paper and therefore I slightly lean towards acceptance (which I reflect in my score). While there may be a scope for a practical comparison on such acclaimed benchmark tasks like MuJoCo (while the reviewer accepts the arguments that as it stands this comparison would be non-trivial), it sways my scores towards acceptance because the topic provides sufficient interest to the research community of safe reinforcement learning. However, given that there is a need to perform substantial revisions in the camera ready version, I am less confident in the final version and therefore reduce my confidence to 3. Also, it would be good to clarify further on the following paragraph in your description:
> > > > > >
> > > > > > "As expected, we observe that the Lagrangian baseline has a high failure rate, particularly in the low-data setting. This makes sense as in the low data setting the concentrability coefficient can be arbitrarily high, and therefore the performance guarantees provided by Le et al. do not hold anymore. As the size of the dataset increases, we observe that the failure rate of the Le et al. starts decreasing, which seems reasonable because with more data more reliable MDP parameters are estimated and the baseline policy now covers more support of the space of all non-stationary policies required for the concentrability assumption to be valid. In contrast, both the MO-SPIBB and MO-SPIBB can ensure low failure rates even in low-data scenarios."
> > > > > >
> > > > > > It would be good to discuss these guarantees of Le et al  (what exact guarantees do not hold anymore? ) a bit more as this will be contributing towards explaining the novelty of the work.

---

> > > > > > > ### Author Response · Authors · 2021-08-31
> > > > > > > **Thank you!**
> > > > > > >
> > > > > > > Thank you for your response and for revising your score. It's much appreciated. We are glad that you believe this work will be of use to the safe RL community.  We provide the clarifications on the comparisons and limitations of the guarantees of Le et al. below.
> > > > > > >
> > > > > > > &nbsp;
> > > > > > >
> > > > > > > The guarantees provided by Le et al. are of the form $\mathcal{J}^\pi_{k,m^{\star}} - \mathcal{J}^{\pi_{b}}_{k, m^{\star}} \geq - \frac{C}{(1-\gamma)^{3/2}}$ (Theorem 4.4 of Le et al.), where $C$ is a term that depends on a constant that comes from the Concentrability assumption (Assumption 1 of Le et al.). This assumption upper bounds the ratio between the future state-action distributions of any non-stationary policy and the baseline policy under which the dataset was generated by some constant. In other words, it makes assumptions on the quality of the data gathered under the baseline policy. Unfortunately, this assumption cannot be verified in practice, and it is unclear how to get a tractable estimate of this constant. As such, this constant can be arbitrarily large (even infinite) when the baseline policy fails to cover the support of all non-stationary policies, for instance, when the baseline policy is not exploratory enough or when the size of the dataset is small. Hence, we observe a high failure rate of Le et al. in the experiments, especially in the low data setting. Compared to Le et al., our performance guarantees do not make any assumptions on the quality of the dataset or the baseline. Therefore, our approach can ensure a low failure rate even in the low-data regime.
> > > > > > >
> > > > > > > &nbsp;
> > > > > > >
> > > > > > > We will take your suggestions into account and highlight this point sufficiently in the draft. Thank you!

---

### Official Review · Reviewer_RGWk · 2021-07-18

**Rating:** 6
**Confidence:** 3

**Summary:**

This paper presents the Safe Policy Iteration with Baseline Bootstrapping (SPIBB) in the offline RL setting with multiple objectives, i.e. multiple rewards. The authors apply the SPIBB set-up to handle the trade-offs for different rewards standing for different user preferences using the multi-objective framework while ensuring that the new policy has the performance guarantee that is at least as well as the behavior policy for each reward function. The authors provide theoretical guarantees of the approach that shows that safe policy improvement is guaranteed for each objective. Finally, the method is evaluated on a grid-world safety task as well as a real-world health care experiment, where the proposed method outperforms previous approaches.

**Limitations And Societal Impact:**

Yes

**Main Review:**

The method is theoretically grounded. The paper is also well written and easy to understand. The experiment results in finite MDPs suggest that the method is both effective in synthetic domains as well as real-world clinical care settings. I think the approach is of great significance to the field of offline RL.

However, I do have a few concerns, which I will discuss as follows.

First, I think the novelty of the paper is somewhat limited. The method seems to be a direct application of SPIBB and HCPI to the multi-objective setting without much modification. The theoretical results seem also to be straightforward extensions to results shown in SPIBB and HCPI, which makes the contribution rather incremental.

Moreover, regarding the experiments, while the authors admit that the results are focused on finite MDPs, I think it is reasonable to apply it to infinite MDPs in empirical evaluations. I would be curious to see how the proposed method can perform in standard offline RL benchmarks such as D4RL where a few domains exhibit the multi-objective structure e.g. antmaze and kitchen. Moreover, I think it would be good to compare the method to previous offline RL methods [1,2,3,4,5,6] combined with vanilla multi-task RL such as shared networks or multi-headed networks. I think vanilla offline MTRL should in principle tackle the proposed problem setting and including such a comparison would be important.

Based on the above comments, I would currently vote for a weak reject.

=================
Post-rebuttal update: After reading the response, I think my concerns are clarified and hence increase my score to a 6.

[1] Kumar,  A.,  Zhou,  A.,  Tucker,  G.,  and Levine,  S.   Conservative q-learning for offline reinforcement learning.Conference on Neural Information Processing Systems,2020.
[2] Ofir Nachum, Bo Dai, Ilya Kostrikov, Yinlam Chow, Lihong Li, and Dale Schuurmans.  Al-gaedice: Policy gradient from arbitrary experience.arXiv preprint arXiv:1912.02074, 2019.
[3] Advantage-weighted regression:Simple and scalable off-policy reinforcement learning.arXiv preprint arXiv:1910.00177, 2019.
[4] Scott Fujimoto, David Meger, and Doina Precup.   Off-policy deep reinforcement learningwithout exploration.arXiv preprint arXiv:1812.02900, 2018.
[5] Aviral Kumar, Justin Fu, Matthew Soh, George Tucker, and Sergey Levine. Stabilizing off-policyq-learning via bootstrapping error reduction.  InAdvances in Neural Information ProcessingSystems, pages 11761–11771, 2019.
[6] Yao Liu, Adith Swaminathan, Alekh Agarwal, and Emma Brunskill. Off-policy policy gradientwith state distribution correction.CoRR, abs/1904.08473, 2019.

**Time Spent Reviewing:**

3 hours

---

> ### Author Response · Authors · 2021-08-10
> **Response to Reviewer RGWk**
>
> Thank you for your feedback and comments. We appreciate you recognizing the significance of the problem, and the thoroughness of our experimental methodology. We respond to your concerns regarding novelty and Deep RL experiments below.
>
> **Novelty/Incremental contributions:**
> Our main contributions in terms of novelty can be broadly summarized as follows:
> 1. We present a novel Seldonian problem formulation that allows the user to experiment with different reward design strategies in safety-critical settings without worrying about the risks of ill-defined scalarizations.
> 2. We show that the existing SPI techniques based on SPIBB and HCPI can be successfully extended to address the problems in this new setting, a result previously unknown to the community. Moreover, we show that the extension of SPIBB to the multi-objective setting also requires us to make additional algorithmic modifications that require incorporating advantage constraints for individual objectives in the optimization problem in order to achieve the requirements regarding practical safety guarantees.
> 3. Finally, the experimental methodology, including the evaluation of the grid world CMDPs and the application of MO-SPIBB and MO-HCPI to the Sepsis study, is an important empirical contribution that will be also useful to any future work in this domain.
>
> **Deep-RL Experiments and Comparisons:**
> The main motivation of our work is to do policy improvement in a safety-critical setting where we can have practical high-confidence guarantees on the performance of the policies returned by our algorithms. Note that this is not the case for the Deep-RL approaches that are currently applied to the Mujoco tasks in the infinite MDPs setting. For instance, none of the works cited by the reviewer [1,2,3,4,5,6] provide practical finite sample high-confidence guarantees nor do they exhibit any empirical evidence on the safety or performance of the deployed policy even in the single-reward setting itself. From a safety-critical viewpoint, this goes against our motivation as we want reliable guarantees on the performance of the policies that we want to eventually deploy. Therefore we believe that a comparison against these works will not be fair. We give a more thorough breakdown of why the works cited by the reviewer fail to address our requirements on practical high-confidence performance guarantees below:
>
> * Works [1, 5] rely on the concentrability assumption, an assumption that cannot be verified in practice.  Concentrability is a strong assumption that upper bounds the ratio between the future state-action distributions of any non-stationary policy and the baseline policy under which the dataset was generated by some constant. From a practical perspective, it is unclear how to get a tractable estimate of this constant, as the space of future state-action distributions of non-stationary policies is vast and as such this assumption can not be verified in practice. Thus, this constant can be arbitrarily huge, potentially even infinite when the baseline policy fails to cover the support of the space of all non-stationary policies, leading to the performance bounds given by these methods to blow up (and even be unbounded).
>
> * In [2] assumptions are made on the future state distributions of the baseline policy under which the dataset is collected, as well as functional assumptions on convexity and the boundedness on the future state distributions of the baseline policy w.r.t. any target policy is made. Additionally, no guarantees on the performance of the policy are made.
>
> * [3] provides no guarantees on the performance of the policy returned by the algorithm.
>
> * In [4], the guarantees are only valid in the deterministic-finite MDPs setting in the scenario where all the possible transitions in the environment are already present in the dataset.  In comparison, we don't restrict to either deterministic MDPs nor do we make any assumptions about the coverage of the dataset, as such assumptions can be unrealistic.
>
> * The work in [6] also makes an assumption on the dataset collected under baseline policy in relation to the optimal policy. In particular, an assumption is made that the baseline policy adequately visits all the states and actions that will be visited by some optimal policy. Additional assumptions related to the smoothness of the structure of the underlying MDP and the returns are also made. There are also no practical performance guarantees.
>
> We failed to show how the existing methods are insufficient for our problem setting. Thank you for pointing this out. We will update the paper to make this point more clear.
>
> Finally, it’s important to note that when it comes to practical application, it is not unusual for continuous domains to be discretized to enable better interpretability, especially when interactions with humans are necessary. For instance, in the Sepsis feasibility study, the domain space is actually high-dimensional where the patient data consists of a 48-dimensional time series with features representing attributes such as demographics, vitals and lab work results (Table 3 in Appendix). A non-parametric function approximation scheme (k-means based clustering) is used to discretize the underlying space. If the Markovian property is valid in the discretized space, the SPIBB-based guarantees will also hold. Thus the finite case is in fact very useful in practice.
>
> We hope that our response alleviates some of your reservations and highlights the importance of the work. We would like to emphasize that even though our formulation is a natural extension, it is technically sound and provides a non-trivial advantage in addressing an important problem. Please let us know if there are any remaining concerns.
>
> References:
>
> [1] Kumar, A., Zhou, A., Tucker, G., and Levine, S. Conservative q-learning for offline reinforcement learning.Conference on Neural Information Processing Systems,2020.
>
> [2] Ofir Nachum, Bo Dai, Ilya Kostrikov, Yinlam Chow, Lihong Li, and Dale Schuurmans. Al-gae dice: Policy gradient from arbitrary experience.arXiv preprint arXiv:1912.02074, 2019.
>
> [3] Advantage-weighted regression:Simple and scalable off-policy reinforcement learning.arXiv preprint arXiv:1910.00177, 2019.
>
> [4] Scott Fujimoto, David Meger, and Doina Precup. Off-policy deep reinforcement learning without exploration.arXiv preprint arXiv:1812.02900, 2018.
>
> [5] Aviral Kumar, Justin Fu, Matthew Soh, George Tucker, and Sergey Levine. Stabilizing off-policy q-learning via bootstrapping error reduction. In Advances in Neural Information Processing Systems, pages 11761–11771, 2019.
>
> [6] Yao Liu, Adith Swaminathan, Alekh Agarwal, and Emma Brunskill. Off-policy policy gradient with state distribution correction.CoRR, abs/1904.08473, 2019.

---

> > ### Author Response · Authors · 2021-08-31
> > **Any further clarifications required?**
> >
> > Dear Reviewer,
> >
> > Thank you again for your valuable feedback that will help us improve this paper.
> >
> > Since the discussion phase of the review period is closing soon, please let us know if our response above has provided the clarifications regarding the submission.
> >
> > Thanks for your consideration and time. It is much appreciated.

---

### Author Response · Authors · 2021-08-24
**Anything else you would like us to respond to?**

Dear reviewers,

Did our response address all your concerns? Do you need more clarifications?

Please, do not hesitate to ask us if additional information is needed.

Thanks for your time!

---

### Author Response · Authors · 2021-08-28
**Le et al. (2019) Comparison Results**

Dear reviewers,

We have completed the comparison to Le et al. (Batch Policy Learning under Constraints, ICML 2019). Thank you for your patience.


**Results:** We test the method by Le et al. (henceforth referred to as Lagrangian) in the synthetic navigation CMDP task described in Section 4 of our draft. We present the results for the best performing Lagrangian baseline on 10 random CMDPs for different $\boldsymbol{\lambda}$ and $\rho$ combinations with $\delta=0.1$ on the anonymous link here: https://upload.vaa.red/i/aTBpa.png. In the figure, each point on the plot denotes the mean (with standard error bars) for 12 different $\lambda$, $\rho$ combinations for the 10 randomly generated CMDPs (120 data points). The x-axis denotes the amount of data the agents were trained on. The y-axis for the left subplot represents the improvement over baseline and the right subplot denotes the failure rate. The dotted black line in the right subplot represents the high-confidence parameter $\delta = 0.1$. The MO-SPIBB is run with $\epsilon = 0.1$ and MO-HCPI with IS = Doubly Robust estimator with student’s t-test concentration inequality. Similar to Figure 1 (a) of our draft, we provide a more detailed plot of how the Lagrangian baseline performs with different hyper-parameters in the above setting here: https://upload.vaa.red/i/rTwJU.png.


As expected, we observe that the Lagrangian baseline has a high failure rate, particularly in the low-data setting. This makes sense as in the low data setting the concentrability coefficient can be arbitrarily high, and therefore the performance guarantees provided by Le et al. do not hold anymore. As the size of the dataset increases, we observe that the failure rate of the Le et al. starts decreasing, which seems reasonable because with more data more reliable MDP parameters are estimated and the baseline policy now covers more support of the space of all non-stationary policies required for the concentrability assumption to be valid. In contrast, both the MO-SPIBB and MO-SPIBB can ensure low failure rates even in low-data scenarios.

**Implementation details and Hyper-parameters:** We build on top of the [publically available code](https://github.com/clvoloshin/constrained_batch_policy_learning) of Le et al. released by the authors and extend it to our setting. The accompanying code for the new experiments can be accessed at the anonymized link here: https://files.catbox.moe/ours2k.zip. In the accompanied code, we also provide a standalone Jupyter notebook ```Lagrange_agent.ipynb``` that contains the implementation of Algorithm 2 of Le et al. (Section 1 of the notebook). We are confident that our implementation is correct as we made sure it passes various sanity tests such as convergence of the primal-dual gap and feasibility on access to true MDP parameters (Section 2 of the notebook).

The algorithm in Le et al (Algorithm 2, Constrained Batch Policy Learning) requires the following hyper-parameters:

* Online Learning Subroutine: We use the same online learning algorithm as used by the authors in their experiments, i.e. Exponentiated Gradient [1].
* Duality gap $\omega$: This denotes the primal-dual gap or the early termination condition. We tried the values in $\set{0.01, 0.001}$ and fix the value to 0.01.
* Number of iterations: This parameter denotes the number of iterations for which the Lagrange coefficients should be updated. We experimented in the range $\set{100, 250, 500}$ and set this to 250.
* Norm bound $B$: The bound on the norm of Lagrange coefficients vector. We tried the values in $\set{1, 10, 50, 100}$ and fixed it 10.
* Learning rate $\eta$: This parameter denotes the learning rate for the update of the Lagrange coefficients via the online learning subroutine. We found that this is the most sensitive variable and we tried with values in $\set{0.005, 0.01, 0.05, 0.1, 0.5, 1.0, 5.0}$. For the final experiments, we benchmark with three different values (0.01, 0.1, 1.0) as mentioned in the second figure we provide above.

We would like to point out that the hyper-parameter tuning for the Lagrangian baseline can be particularly challenging as in the low-data setting none of the combinations of the above hyper-parameters can ensure a low failure rate even though the duality gap has converged. We provide an example of this phenomenon in Section 3 of the accompanying notebook.


**Environments in Le et al.:** Le et al. test their approach on two domains: a grid-world domain under safety constraint, and a high-dimensional car racing domain. The car racing domain takes the raw pixel image tensor as input, and as we mentioned in the limitations, it is out of the scope of our work. We would like to highlight that the grid-world domain and empirical methodology in Le et al. are considerably weaker than the approach we take in our work with respect to the safety constraints. This can be observed in Figure 2 (middle) of Le et al. where even the online-RL (equivalent to the Linearized baseline in our case) also has no constraint violation. Moreover, they do not experiment with the different sizes of the dataset $(|\mathcal{D}|)$ or the quality of the baseline under which the dataset was collected $(\rho)$. Compared to that, we base our grid-world environments on the standard CMDP safety benchmarks [2,3], have comparisons against different dataset sizes and baseline quality parameters, and also explicitly calculate the failure rate.

&nbsp;

The above experiments show the advantage of our approach over Le et al., particularly in the low-data safety-critical tasks, where our methods can improve over the baseline policy while ensuring a low failure rate. Please let us know if you have any remaining questions.

&nbsp;


References:

[1] Kivinen, J. and Warmuth, M. K. Exponentiated gradient versus gradient descent for linear predictors. Information and Computation, 132(1):1–63, 1997.
[2] ​Leike, J., Martic, M., Krakovna, V., Ortega, P. A., Everitt, T., Lefrancq, A., Orseau, L., and Legg, S. (2017). AI safety grid worlds. arXiv preprint arXiv:1711.09883.
[3] Chow, Y., Nachum, O., Duenez-Guzman, E., and Ghavamzadeh, M. (2018). A Lyapunov-based approach to safe reinforcement learning. arXiv preprint arXiv:1805.07708.

---

### Decision · Program_Chairs · 2021-09-27

**Decision:**

Accept (Poster)

**Comment:**

The paper proposed a SPIBB algorithm for offline constraint RL. This is an interesting problem to address. Most of the reviewers think the theoretical analysis is rather weak in the paper, but the experiments give good complementation to the theory. Hence the overall recommendation is a weak acceptance.